# Reproductive performance of lumpfish (*Cyclopterus lumpus*, L. 1758) females: Effects of integrated photoperiod and temperature manipulations on sexual maturation and spawning

Frank Thomas Mlingi [1¤a] *, Erik Burgerhout[2], Maren Mommens[3¤b], Helge Tveiten[4], Jonna Tomkiewicz [5], Elin Kjørsvik [1], Velmurugu Puvanendran [2]

1 Department of Biology, Norwegian University of Science and Technology (NTNU), Trondheim, Norway, 2 Department of Production Biology, Nofima AS, Tromsø, Norway, 3 Department of Breeding and Research and Development, AquaGen AS, Trondheim, Norway, 4 Norwegian College of Fisheries Science, UiT The Arctic University of Norway, Tromsø, Norway, 5 National Institute of Aquatic Resources, Technical University of Denmark, Kongens Lyngby, Denmark

¤a Current address: Ode As, Stadsbygd, Norway
¤b Current address: Thelma Biotel AS, Trondheim, Norway
* frank.t.mlingi@ntnu.no, frank.mlingi@ode.no

## Abstract

A successful control of sexual maturation is crucial for year-round production of lumpfish juveniles destined as cleaner fish in Atlantic salmon aquaculture. This study investigated the combined effects of photoperiod and temperature manipulations on sexual maturation and spawning in lumpfish females. Lumpfish juveniles were exposed to simulated natural and nine-month compressed annual photoperiods, with subsequent temperature elevation. Body weight (BW), condition factor (K), gonadosomatic index (GSI), ovarian development, plasma levels of 17β-estradiol (E2), testosterone (T) and 11-ketotestosterone (11-KT), and spawning were assessed. Compressing the natural photoperiod caused a clear increase and decrease in GSI, T, 11-KT and E2 towards and during the spawning period. Before the temperature elevation, GSI, T, 11-KT, E2 and ovarian development were advanced in the compressed photoperiod. After the temperature elevation, GSI, T, 11-KT and E2 fluctuated more in the compressed photoperiod, while in the natural photoperiod, E2 declined, and GSI, T and 11-KT increased. Spawning was advanced by 1 month in the compressed photoperiod compared to the natural photoperiod. Temperature elevation led to higher levels, earlier peaks and declines of T, 11-KT or E2 in both photoperiods, and advanced spawning by 1.5 months in the compressed photoperiod compared to the natural photoperiod. Temperature elevation also led to increased ovulation recruitment and increased cumulative weight of spawned eggs in the natural photoperiod. Compressing the natural photoperiod and elevating temperature can thus advance sexual maturation and spawning in lumpfish females. Due to the lower amounts of spawned egg weights in the high temperature compressed photoperiod, further studies on effects of photoperiod and timing of temperature

**Data Availability Statement:** All relevant data are within the manuscript and its Supporting information files.

**Funding:** This work is part of a research project titled "Reproductive biology of lumpfish (Cyclopterus lumpus): a key to successful selective breeding (CycloBreed), which was funded by the Norwegian Seafood Research Fund (Fiskeri-og havbruksnæringens forskningsfond (FHF))) with grant number 901418. The funder did not play any role in the study design, data collection and analysis, decision to publish, or preparation of the manuscript.

**Competing interests:** The authors have declared that no competing interests exist.

manipulations on spawning, fecundity and egg quality could optimize the photothermal manipulations on lumpfish broodstock.

## Introduction

Lumpfish (*Cyclopterus lumpus*) is increasingly used as a cold-water alternative biological control against sea lice infestations in Atlantic salmon (*Salmo salar*) cages [1–3]. Hence the number of lumpfish hatcheries that produce lumpfish juveniles is growing to sufficiently supply the high demand [4, 5]. The increased production and high demand are due to challenges associated with the use of other methods of sea lice treatment and more positive perceptions from consumers and environmentalists on the use of lumpfish as a cleaner fish [6].

The increase in juvenile production occurs despite the insufficient knowledge on lumpfish reproductive biology [2, 6]. This production is largely dependent on the capture of wild broodstock for eggs and sperm during the spawning season [2]. However, the extensive spawning season of lumpfish [7, 8] cannot ensure a consistent juvenile production, as there are variable factors such as temperature, which cause variations in egg quality [8]. Also, there is a risk of introducing diseases from the wild while using wild-caught broodstock [2]. Successful control of sexual maturation is therefore, crucial for sustainably supplying adequate and right-sized juveniles for stocking into salmon cages year-round andreducing fishing pressures on wild populations [2]. With a sustainable hatchery production of lumpfish juveniles, a year-round supply that matches the natural dynamic sea lice occurrences and the sea transfer of smolts can be achieved, and an optimal utilization of the production facilities will be ensured [9].

Furthermore, variations between families, such as growth, feeding behavior and sea lice grazing efficacy are important considerations in the production of lumpfish juveniles destined to serve as cleaner fish [10]. Therefore, targeted family production, along with selection and breeding programs aimed at enhancing traits such as sea lice grazing, feeding behavior, and reducing size variation and cataract prevalence, are crucial for producing robust lumpfish juveniles [10, 11]. In Norway, there has been a gradual transition from using wild-caught broodstock to intensive farming of lumpfish, with a lumpfish breeding program was established in 2017 [12]. To ensure the production of robust lumpfish juveniles for cleaner fish purposes, family lines with desired traits should originate from targeted breeding programs [2]. For the establishment of successful breeding programs, closing the life cycle in captivity is essential [2]. This can be achieved through environmental manipulations, which will make the timing of lumpfish juvenile production more predictable. Consequently, a predictable production of juveniles will enhance the sustainability and efficiency of breeding program design and implementation.

In temperate regions, photoperiod and temperature regulate the fish reproductive cycle, and these fishes have a seasonality adaptation for reproduction [13, 14]. Their developmental and maturational events are synchronized with changes in climate, day length, temperature and larval food availability with the primary objective of ensuring survival of the offspring [15]. The seasonal variations in photoperiod are argued to be the most proximate cue responsible for the signaling and timing of reproduction in the majority fish species [15]. However, temperature influences the reproductive development, and often acts secondarily to synchronize the final stages of sexual maturation with factors that are necessary for the offspring survival [14–16].

In captivity, photoperiod and temperature are manipulated for the control of sexual maturation in various fish species to ensure year-round availability of juveniles [13, 17]. However, due to specificities in reproductive patterns, defining a species-specific optimal photothermal regime is crucial [13, 18]. In lumpfish, it has been shown that, photoperiod can alter the spawning time, egg volume, and hatching success [9, 19]. On the other hand, knowledge on the role of temperature is limited to spawning activity and egg viability, where relatively high temperature (i.e., 14°C) caused negative effects, under a constant photoperiod [8].

Although manipulation of photoperiod warrants the potential for a successful control of sexual maturation in lumpfish, the role of temperature as an important secondary cue is not well understood. To this end, the control of sexual maturation and the subsequent spawning through photothermal manipulations to ensure a sustainable year-round production of lumpfish juveniles has not been fully achieved. Recently, we conducted a short-term exposure of lumpfish broodstock to either a continuous or a short-to-continuous photoperiod combined with temperature elevation during final maturation. We found that the short-to-continuous photoperiod possessed the potential to synchronize and advance sexual maturation, and that temperature elevation modulated the photoperiod effects of sexual maturation. To understand long-term photothermal effects, we designed an experiment starting with immature lumpfish juveniles. We aimed at exposing lumpfish juveniles to simulated natural and nine-month compressed annual photoperiods. With reference from the increase in gonadosomatic index and ovarian development, temperature elevation was conducted. We hypothesized that compressing the natural photoperiod would advance sexual maturation and spawning, and that temperature elevation would enhance the photoperiod effects of the reproductive performance.

## Materials and methods

### Animal welfare statement

The experiment, all fish handling, experimental design, and procedures were approved by the Norwegian Food Safety Authority (Mattilsynet with FOTS ID: 12164). The fish were reared and maintained in accordance with the Norwegian Animal Welfare Act of 20th December 1974, No. 73, Sections 20–22, amended on 19th June 2009. For sampling, all fish were euthanized using an overdose of tricaine methane sulfonate (165 mg L$^{-1}$, MS222, Finquel, Argent Chemical Laboratories Inc., Redmond, USA).

### Animals, experimental design, and sampling

The experiment was carried out from 30th January 2018 to 1st June 2019, at the Center for Marine Aquaculture in Kraknes, Tromsø, Norway. Originating from eggs fertilized with sperm, both from wild broodstock, immature lumpfish juveniles aged 12 months, previously reared under a continuous photoperiod (L:D = 24:0) at 10°C, were obtained.

A total of 4000 individuals (mixed sex, body weight (BW) and body length (BL): 157.2 ± 32.2 g and 14.9 ± 1.1 cm (average ± SD), respectively) were transferred into four 10 m$^{-3}$ tanks at a density of 1000 fish/tank (100 fish m$^{-3}$). At ambient temperature (0T), two of the four tanks were exposed to a natural annual photoperiod (NP, hence NP0T) while the other two tanks were subjected to a nine month-compressed annual photoperiod (CP, hence CP0T).

The timing of temperature elevation was dependent of the maturation status, which was assessed based on gonadosomatic index (GSI) in females, and the onset of final oocyte maturation. Consequently, from 3rd January 2019 and 6th March 2019, in CP0T (receiving approximately 18 hours of light) and NP0T (receiving 13 hours of light), respectively, the temperature in one tank was elevated until it reached approximately 3°C higher than the ambient temperature, after approximately two days. On 3rd January 2019, GSI in CP was 7.5 ± 4.5 and in NP it

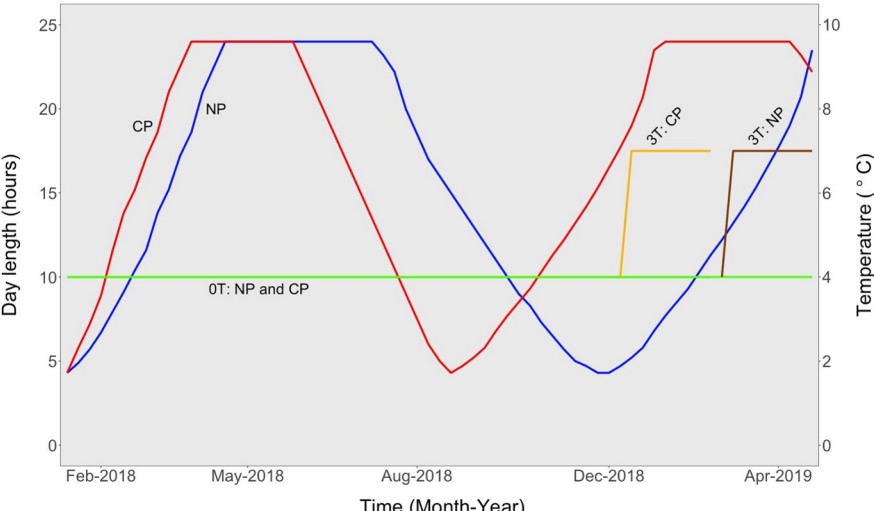

**Fig 1. Experimental setup.** Experimental setup showing the natural and compressed photoperiods, and the different timings of temperature elevation. CP = compressed photoperiod, NP = natural photoperiod, 0T = ambient temperature, 3T = elevated temperature.

was $3.8 \pm 2.2$ (mean ± SD). The temperature elevation resulted in four treatments: NP0T (natural photoperiod at ambient temperature), NP3T (natural photoperiod at elevated temperature), CP0T (compressed photoperiod at ambient temperature) and CP3T (compressed photoperiod at elevated temperature) (Fig 1).

Sampling was conducted at the start of the experiment, on average once every 34 days before temperature elevation and, once every 14 days after the first temperature elevation. Initially, five females were sampled, and thereafter, at least eight females were sampled per tank throughout the rest of the experimental period.

Body weight (BW) (g) and body length (BL) (cm) were measured and used to obtain Fulton's condition factor as: $K = (BW/BL^3) \times 100$.

Blood was collected from the caudal vessels into vacutainer tubes (BD Vacutainer LH 68 I. U. Plymouth, UK). The blood was centrifuged at 5000 rpm at 4°C for 10 minutes to extract blood plasma. The blood plasma was stored at -20°C until further analyses.

Gonads were excised from the fish, and gonad weight (GW) was measured to calculate gonadosomatic index (GSI) as: GSI (%) = (gonad weight / total body weight) * 100. Gonad tissues were then cut and placed in histology cassettes (Simport Histonette Tissue Processing/ Embedding Cassettes with Lid), fixed in 4% buffered formalin and, stored at 4°C until histological examination for gonadal development.

After the temperature elevation, monitoring of spawning was initiated, where observations of color changes in males and plumpness in females were used as indications of readiness to spawn. During the active spawning period, each tank was checked for eggs every morning, and all separate egg clumps (batches) were collected, counted and weighed.

Throughout the experimental period, temperature (°C) and oxygen (% saturation) were monitored daily (see S1 Fig).

## Quantification of ovarian development

The processing of ovarian tissues, identification and quantification of oogenesis stages were conducted as described in a previous work [20]. The establishment of ovarian development

stages was done with reference to proposed standard terminologies: primary growth (PG), secondary growth (SG), final/oocyte maturation (OM) and ovulation (OV) [21, 22].

## Sex steroid analyses

Plasma levels of 17β-estradiol (E2), testosterone (T), and 11-ketotestosterone (11-KT) were measured using Radioimmunoassay (RIA), following established protocols [23]. The assay characteristics and cross-reactivities of E2 and T antisera (isolated from New Zealand White (NZW) rabbits) were assessed by Frantzen and colleagues [24], while cross-reactivities of the 11-KT antiserum (isolated from NZW rabbits) were evaluated by Johnsen and colleagues [25].

To extract free steroids, the blood plasma was mixed with diethyl ether (DEE) and subjected to vigorous shaking. The steroids were obtained after phase separation, evaporation of DEE and reconstitution using RIA-buffer. During RIA, the extracts were mixed with the specific tracer and antiserum. The mixture was incubated and later subjected to charcoal extraction. Radioactivity was then measured by scintillation counting. From pooled plasma, dilution curves parallel to the standard curves of different steroid assays were obtained. Values below the detection limits (0.2 ng/ml) were assigned a value equal to half of the detection limit (0.1 ng/ml). Details of the extraction and RIA are described in a previous work [20].

## Confounders

Due to increased mortalities (see S3 Fig), on 17[th] October 2018, samples from four (4) dead fish were taken for identification of the nature of the symptoms. These fish were diagnosed with atypical furunculosis, and a 10-day medication through feed was initiated on 22[nd] October 2018. Although this flavivirus infection was not our primary exposure of interest, it significantly contributed to imbalanced sampling due to high fish mortalities. While acknowledging the potential effects of this condition on our results, it's important to note that the possible confounding effects of the infection were not specifically analyzed in this study.

## Statistical analyses

The data visualizations and statistical analyses were conducted using RStudio (version 1.4.1106) for R (version 4.0.4), supported by the packages 'tidyverse', 'car', 'MASS', 'lme4' and 'emmeans' [26–29]. All data are presented as Mean ± Standard Error of the Mean (Mean ± S.E.M.) unless stated otherwise.

The effects of photoperiod, temperature and time on BW, K, GSI, and plasma levels of T, 11-KT and E2 were analyzed. Before temperature elevation (from 30[th] January 2018 to 3[rd] January 2019), the effect of photoperiod (NP0T and CP0T) and the change over time, along with their interactions were analyzed. Following temperature elevation (from 3[rd] January to 24[th] April 2019) sampling was not uniform. The analyses were therefore organized into several categories as shown in Table 1.

Normality and homoscedasticity were assessed using Shapiro-Wilk and Levene's tests, respectively. Data with normal distribution and homoscedasticity were analyzed by fitting linear mixed effects models, with tanks treated as nested random effects. The data with non-normal distribution and/or heteroscedasticity were subjected to boxcox transformation to determine a lambda value with the highest log-likelihood. Subsequently, generalized linear mixed effects models with Gamma errors and power link function were fitted, with tanks as nested random effects.

Post hoc tests were conducted using Tukey's honest significance tests, with P values of < 0.05 considered significant.

**Table 1. Sampling points selected for statistical analyses after the first temperature elevation.** Groups which were sampled together are shaded with the same color, and sampling points for these groups are marked by "**X** ".

| Sampling points | Groups | | | | | | | | | | |
|---|---|---|---|---|---|---|---|---|---|---|---|
| | NP0T | NP3T | CP0T | CP3T | NP0T | CP0T | CP3T | NP0T | NP3T | CP0T | CP3T |
| 03/01/2019 | X | X | X | X | X | X | X | X | X | X | X |
| 17/01/2019 | | | | | | | | | | X | X |
| 31/01/2019 | | | | | X | X | X | | | X | X |
| 14/02/2019 | | | | | | | | | | X | X |
| 28/02/2019 | | | | | X | X | X | | | X | X |
| 14/03/2019 | X | X | X | X | X | X | X | X | X | X | X |
| 27/03/2019 | | | | | | | | X | X | | |
| 10/04/2019 | | | | | | | | X | X | | |
| 24/04/2019 | | | | | | | | X | X | | |

The materials and methods are also summarized in a flowchart (Fig 2) which presents the entire process from experimental setup to data processing.

## Results

### Somatic growth, ovarian growth and development, and spawning

**Body weight and condition factor.** There was a steady and similar increase in mean body weight (BW) from the start of the experiment prior the temperature elevation in CP. After the temperature elevation in the CP (from 3rd January 2019), the mean BW fluctuated within the different groups, although generally slightly increasing over time. In CP3T, BW increased until 14th February 2018 and then decreased afterwards, while in CP0T, BW fluctuated reaching it's highest value on 14th March 2019. BW was slightly higher in CP0T compared to CP3T and NP0T until the final sampling in CP on 14th March 2019. After the temperature elevation in the NP (from 6th March 2019), the mean BW in NP groups showed fluctuations, with an upward trend towards 24th April 2019. These fluctuations were less pronounced in NP3T compared to NP0T. During this period, BW in NP0T was slightly higher than in NP3T. No significant differences in mean body weights among groups were observed (Fig 3).

Independent of temperature elevation, K showed slight fluctuations across all groups (see S2 Fig). Following the temperature elevation, there was a general slight decline, which appeared to be more pronounced in the CP groups. In both photoperiods, the decline was more noticeable in the high temperature groups compared to the ambient temperature groups. Additionally, there were surges in K in CP0T on 28th February and in NP0T 14th March 2019. No significant differences in mean condition factor were observed among the groups.

*Gonadosomatic index (GSI).* There was a gradual increase in GSI before temperature elevation in both NP0T and CP0T. GSI was significantly higher in CP0T on 3rd January 2019 before the start of temperature elevation. After the temperature elevation in CP, mean GSI increased more rapidly in CP3T than in CP0T, and maximum GSI occurred earlier in CP3T than in CP0T.

After the temperature elevation in NP on 6th March 2019, mean GSI in both NP groups increased to maximum levels in April. The CP groups exhibited slightly higher GSI than the NP groups until their (CP) last sampling on 14th March 2019 (Fig 3). Notably, changes in GSI were advanced in the elevated temperature groups, where GSI was slightly higher in the beginning, but later declined to becoming slightly lower than in the ambient temperature groups. Peaks in GSI were reached earlier in the CP groups in the order: CP3T (14th February 2019),

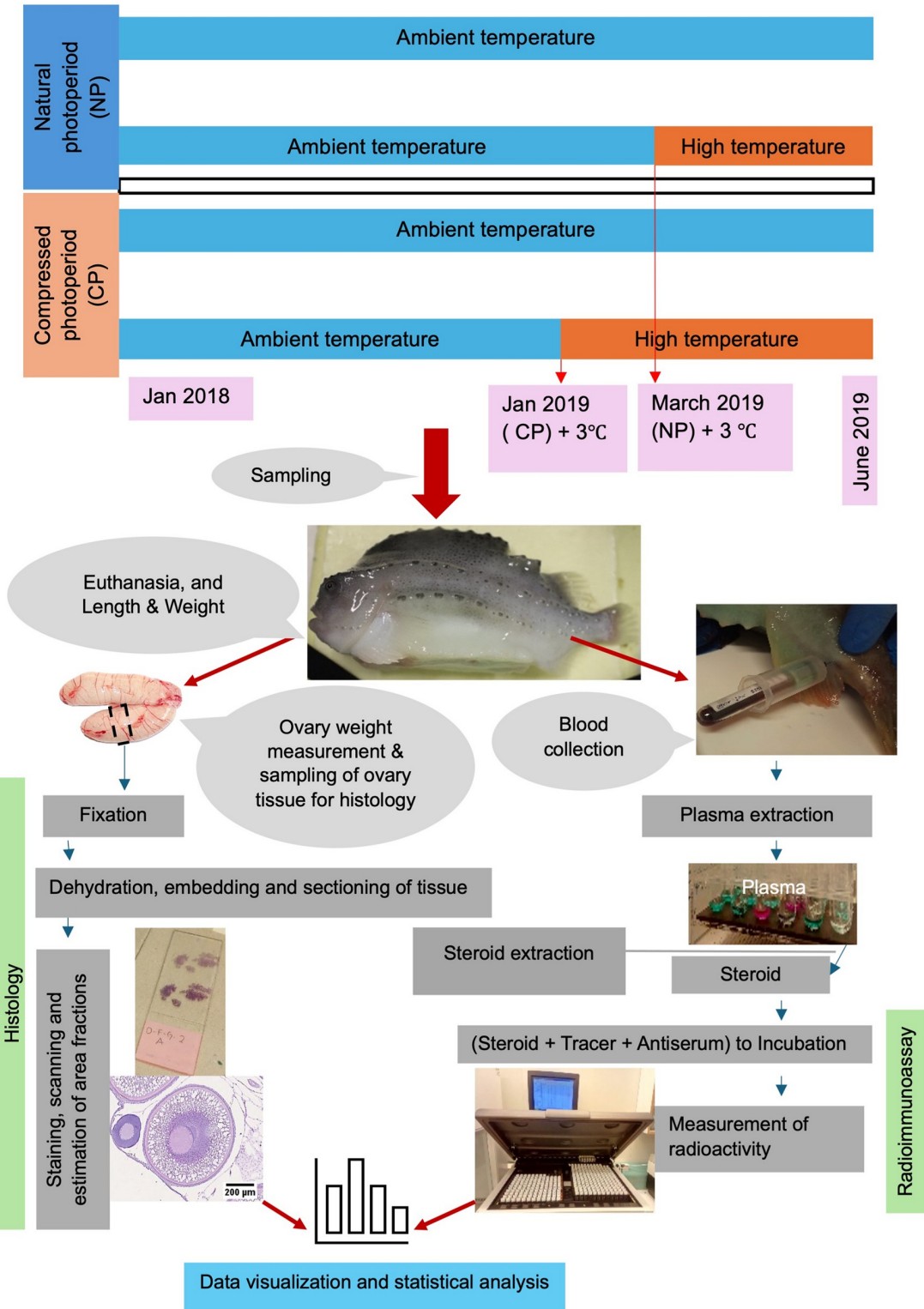

**Fig 2. Flow chart presenting the experimental setup, sampling and steps of histology and sex steroid data collection for assessment of the lumpfish female reproductive performance.** The timing of temperature elevation (+ 3˚C) was based on the maturation status in terms of gonadosomatic index and onset of final oocyte maturation.

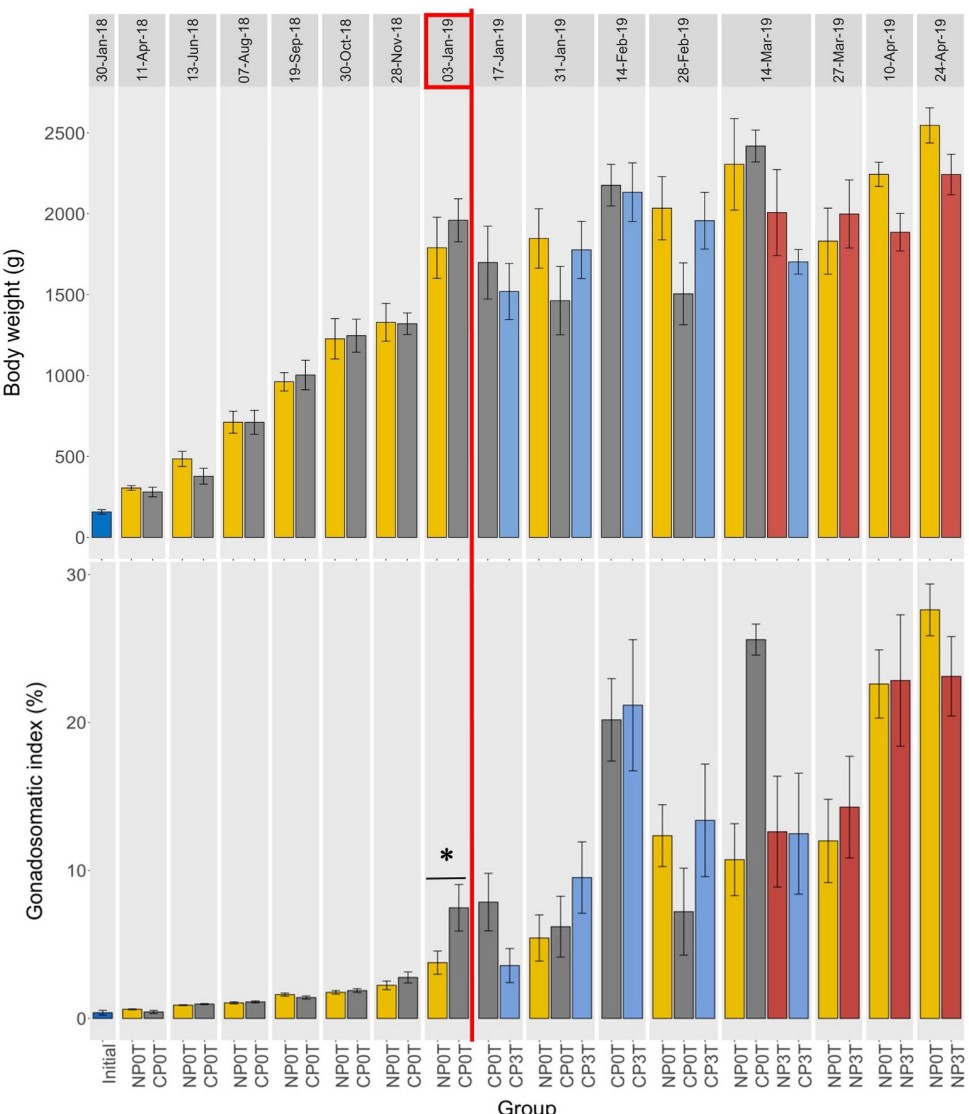

**Fig 3. Photoperiod and temperature effects on body weight (top) and gonadosomatic index (bottom).** Body weights and gonadosomatic indices of lumpfish females from different photoperiods (until 03-Jan-19), and different photoperiod and temperature combinations (from 03-Jan-19). NP0T: natural photoperiod at ambient temperature, CP0T: compressed photoperiod at ambient temperature, NP3T: natural photoperiod at elevated temperature, CP3T: compressed photoperiod at ambient temperature. The red rectangle on 03-Jan-2019 indicates the start of temperature elevation, and the red vertical line separates the periods before and after temperature elevation. Values are mean ± S.E. M and statistical significances between groups at a sampling point are indicated by "*" or "ns". Statistical differences between sampling points are graphically displayed in supporting information on S8–S14 Figs. For trend visualization, refer to (S4 Fig), which presents the data as a line graph.

CP0T (14[th] March 2019), and NP3T/NP0T (24[th] April 2019) (Figs 6 and 7). No significant differences were found between groups at any sampling points after temperature elevation.

*Ovarian development*. The primary growth (PG) stage was present throughout, with a temporal decreasing trend towards the end of the experiment. Minimal recruitment of secondary growth (SG) was observed for the first time on 13[th] June 2018 (when both groups were receiving approximately 24 hours of light) only in NP0T. On 7[th] August 2018 (when CP0T was

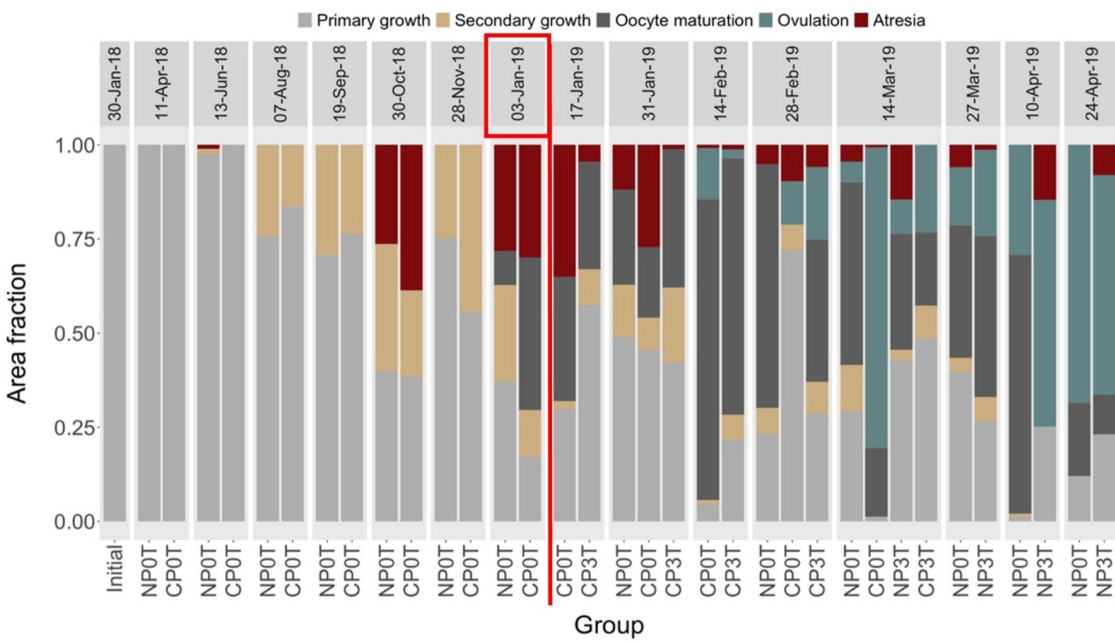

**Fig 4. Photoperiod and temperature effects on ovarian development.** Mean area fractions (in proportions) of ovarian stages in lumpfish females from different photoperiods (until 03-Jan-19), and different photoperiod and temperature combinations (from 03-Jan-19). NP0T: natural photoperiod at ambient temperature, CP0T: compressed photoperiod at ambient temperature, NP3T: natural photoperiod at elevated temperature, CP3T: compressed photoperiod at ambient temperature. The red rectangle on 03-Jan-2019 indicates the start of temperature elevation, and the red vertical line separates the periods before and after temperature elevation.

under 11–12 light hours and NP0T under 22–23 light hours), the SG stage was observed in both NP0T and CP0T, with a larger area fraction in NP0T. A decreasing trend in the fractions of SG in all groups was observed until 10th April 2019, after which SG was no longer observed. The oocyte maturation (OM) stage was first observed on 3rd January 2019 (when CP was under 17–18 light hours and NP under 4–5 light hours), with CP0T containing a larger area fraction compared to NP0T. Recruitment of OM continued in varying area fractions among the groups until 10th April 2019 and decreased afterward. The ovulation (OV) stage was first observed in both CP groups on 14th February 2019, six weeks after the first temperature elevation (when CP was under 24 light hours and NP under 8–9 light hours). The area fractions of the OV stage were low in these groups throughout February. On 14th March 2019, the OV stage in CP0T was > 80%, whereas in CP3T it remained low.

On 14th March 2019, about one week after the temperature elevation in NP, small area fractions of OV were observed in both NP groups. However, OV recruitment was mostly higher in NP3T than in NP0T, but on 24th April 2019 both groups showed > 60–70% area fractions of the OV stage. Atretic (ATR) oocytes appeared in all groups, especially between 30th October 2018 and 31st January 2019, which corresponded with the period before OV. ATR was observed in 75% of the sampling points, with higher proportions observed before than during the spawning period (Fig 4).

## Sex steroids

**Testosterone (T) and 11-Ketotestosterone (11-KT).** Before temperature elevation, T and 11-KT levels were low in both NP0T and CP0T. After temperature elevation, an increase

occurred towards the spawning period (from 4th February 2019), and this rise was observed earlier in CP0T. At the start of the spawning (4th February 2019), the highest mean plasma levels of T and 11-KT were reached, and these were in CP3T. This was followed by a decline that was linear in CP3T compared to CP0T. While T and 11-KT levels in NP0T increased slightly and linearly, in NP3T, there was an increase until 27th March 2019 followed by a decrease where T became significantly lower than in NP0T (Fig 5). Notably, the T and 11-KT changes were more advanced in the high-temperature groups compared to the ambient temperature groups. Peaks in T and 11-KT occurred earlier in the CP groups: For T, peaks were on 14th February 2019 for CP3T and CP0T, and 27th March 2019 for NP3T and NP0T. For 11-KT, peaks were on 14th February 2019 for CP3T and CP0T, 27th March 2019 for NP0T and 24th April 2019 for NP3T (Figs 6 and 7).

Significant differences were observed in T levels: mean T in CP0T was significantly higher than in NP0T on 3rd January 2019, in CP3T, mean T was significantly higher than in NP0T on 28th February 2019, and mean T in NP0T was significantly higher than in NP3T on 10th April 2019. Similarly, significant differences were noted in 11-KT levels: mean 11-KT in CP0T was significantly higher than in NP0T on both 3rd January and 14th March, 2019, and in CP0T, mean 11-KT was significantly higher than in CP3T on 14th March 2019 (Fig 5).

**Estradiol (E2).**   Overall, there was an increase in E2 levels towards the spawning period (from 4th February 2019), followed by a linear decrease until the end of the experiment on 24th April 2019. Notably, compared to T and 11-KT, the increase in E2 occurred earlier, starting from 19th September 2018. E2 levels in CP3T, CP0T, NP0T and NP3T reached their peaks on 31st January, 14th February, 28th February, and 14th March 2019, respectively, and decreased afterwards, with the decline occurring earlier in the CP groups. Regarding temperature, the high-temperature groups exhibited a faster decrease in E2 levels. Specifically, in NP3T, E2 dropped from its peak on 14th March 2019, becoming significantly lower than in NP0T on 24th April 2019 (Fig 5).

*Spawning.* Spawning events were observed earliest in CP3T from 4th February 2019, followed by CP0T from 14th February 2019, NP0T from 8th March 2019, and lastly NP3T from 14th March 2019. Spawning lasted for 2 months, 3 months, 2 months, and 7 weeks in CP3T, CP0T, NP0T, and NP3T, respectively. The peak in spawning activities was reached earliest in CP3T in February 2019, followed by CP0T in March 2019, and lastly NP3T and NP0T in April 2019. Cumulatively, the spawned egg weight was larger in CP3T than in CP0T initially, but later, the spawned egg weight in CP0T exceeded that in CP3T, progressively becoming the largest throughout the rest of the egg collection period. The cumulative egg weight in NP0T was initially larger than in NP3T but later it was exceeded and, NP3T was second after CP0T throughout the remaining egg collection period (Fig 8). A summary of spawning and peak timings of GSI, T, 11-KT and E2 is presented in Fig 9.

## Discussion

### Body weight and condition factor

Body weight increased and exhibited fluctuations after temperature elevation that were more advanced in the compressed photoperiod. Similar findings were reported by Imsland and colleagues [19] who observed changes in growth patterns of lumpfish females, affecting the age at first maturity, following exposure to compressed annual photoperiods. Fluctuations in body weight during the spawning period could be attributed to feeding suppression, a phenomenon observed also in the Atlantic cod [30], likely more intensively in the high-temperature groups. The trade-off of energy resulting in negative effects on somatic growth over reproduction, as observed in the Atlantic cod [31] could also be possible.

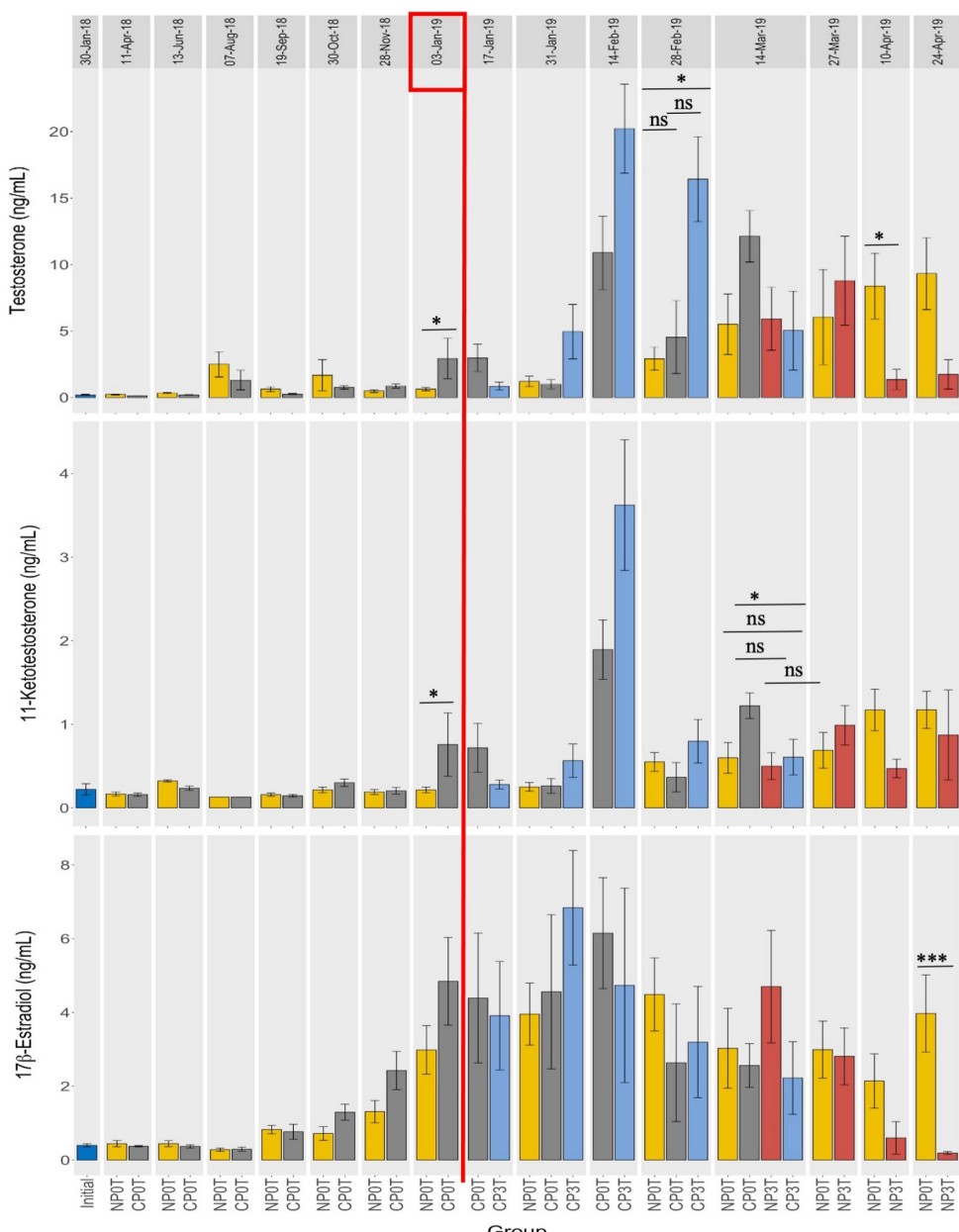

**Fig 5. Photoperiod and temperature effects on sex steroids.** Blood plasma levels of testosterone (top), 11-ketotestosterone (middle) and 17β-estradiol (bottom) in lumpfish females from different photoperiods (until 03-Jan-19), and different photoperiod and temperature combinations (from 03-Jan-19). NP0T: natural photoperiod at ambient temperature, CP0T: compressed photoperiod at ambient temperature, NP3T: natural photoperiod at elevated temperature, CP3T: compressed photoperiod at ambient temperature. The red rectangle on 03-Jan-2019 indicates the start of temperature elevation, and the red vertical line separates the periods before and after temperature elevation. Values are mean ± S.E.M and statistical significances between groups at a sampling point are indicated by "*" or "ns". Statistical differences between sampling points are graphically displayed in supporting information on S8–S14 Figs. For trend visualization, refer to (S5 Fig), which presents the data as a line graph.

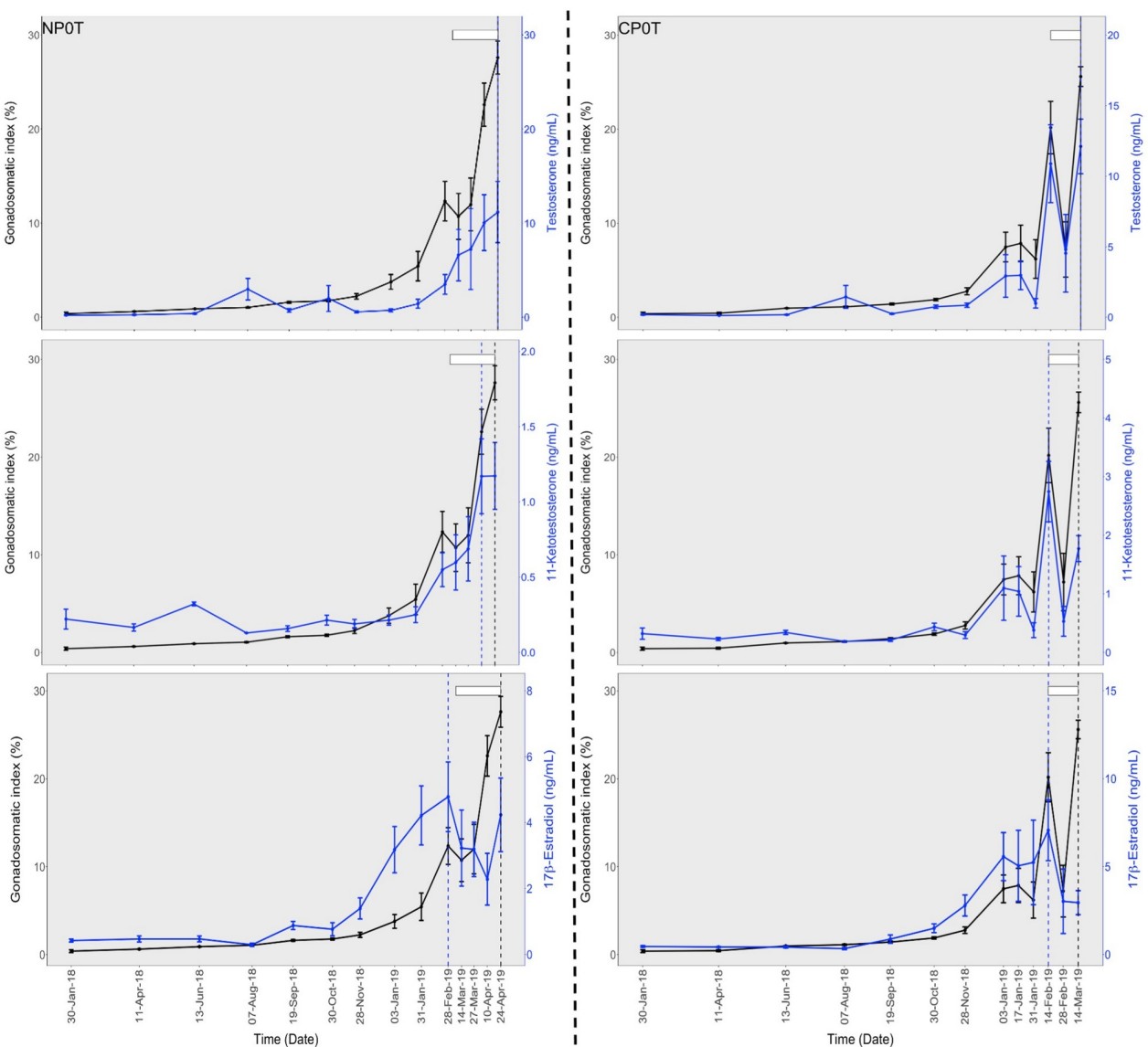

**Fig 6.** Gonadosomatic index and sex steroid profiles, and timing of spawning under natural (left) and compressed (right) photoperiods at ambient temperatures. Temporal changes in gonadosomatic index (GSI) and testosterone (top), 11-ketotestosterone (middle) and 17β-estradiol (bottom) in lumpfish (Cyclopterus lumpus) females exposed to natural and compressed photoperiods at ambient temperatures (NP0T and CP0T). The white bars indicate the timing of spawning (from start of spawning to the end of sampling/experiment), the vertical broken lines indicate the timing of peaks in GSI (black) and sex steroids (blue). The time breaks are based on the sampling intervals. Values are mean ± S.E.M and the statistical differences are graphically displayed in supporting information on S8–S14 Figs.

The decline in condition factor during the spawning period is attributed to the spawning activities, consistent with reports in lumpfish elsewhere [32, 33]. The seemingly earlier decline in condition factor in the compressed annual photoperiod is likely because of the effect of the photoperiod regime on sexual maturation and, consequently, earlier spawning. This aligns with a previous study on lumpfish females, where the condition factor started declining significantly in a compressed photoperiod compared to a natural photoperiod [19]. The secondary effect of temperature on sexual maturation and eventual spawning is also possible, as condition

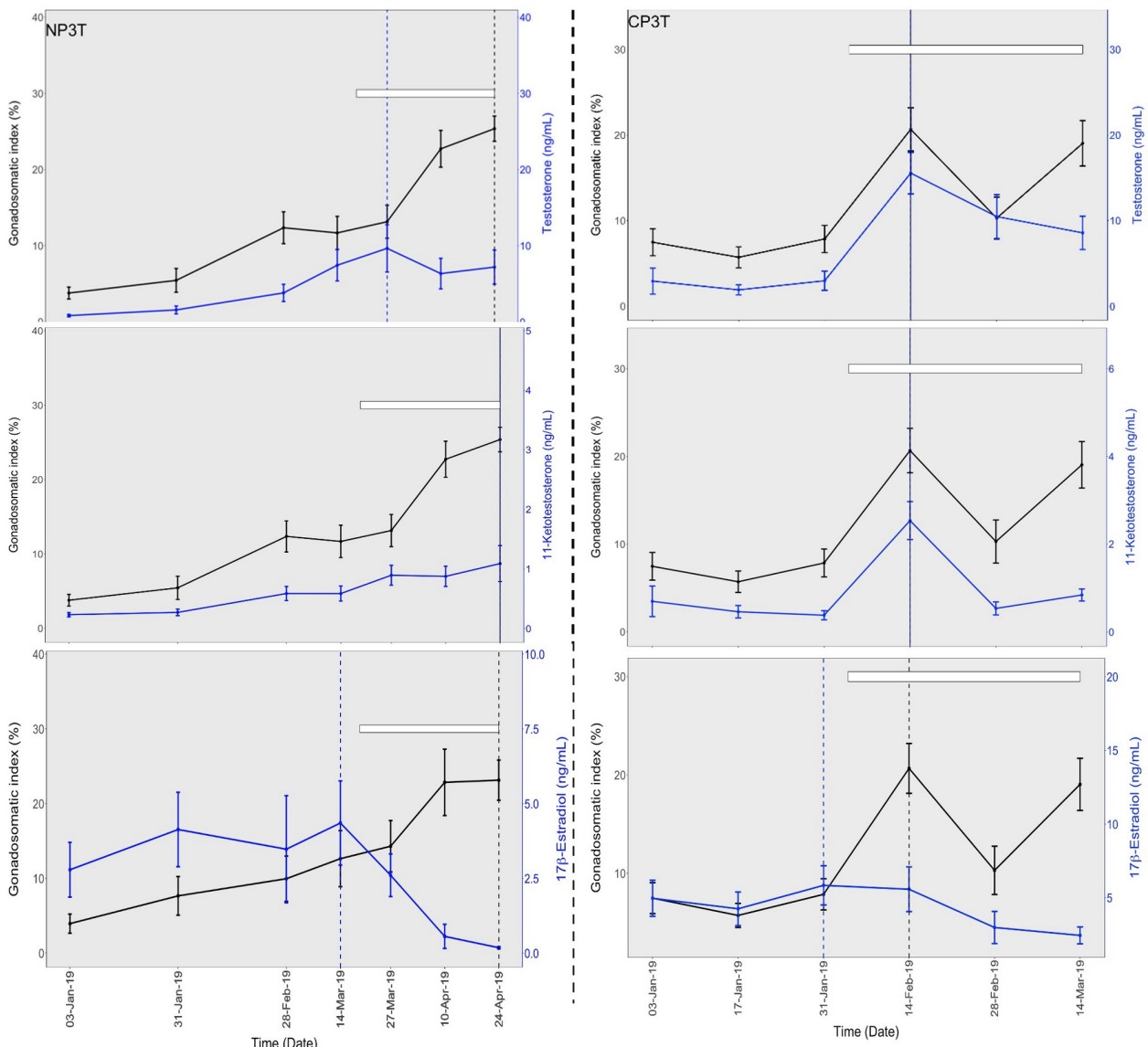

**Fig 7.** Gonadosomatic index and sex steroid profiles, and timing of spawning under natural (left) and compressed (right) photoperiods at elevated temperatures. Temporal changes in gonadosomatic index (GSI) and testosterone (top), 11-ketotestosterone (middle) and 17β-estradiol (bottom) in lumpfish (Cyclopterus lumpus) females exposed to natural and compressed photoperiods at elevated temperatures (NP3T and CP3T). The white bars indicate the timing of spawning (from start of spawning to the end of sampling/experiment), the vertical broken lines indicate the timing of peaks in GSI (black) and sex steroids (blue). The time breaks are based on the sampling intervals. The red vertical line separates the periods before and after temperature elevation. Values are mean ± S.E.M and the statistical differences are graphically displayed in supporting information on S8–S14 Figs.

factor in the elevated temperature groups seemed to decline slightly faster compared to those in the ambient temperature groups.

## Sexual maturation and spawning

Throughout the experiment, the gonadosomatic index (GSI) and sex steroids increased towards the spawning period (starting from 4th February 2019) and decreased later, especially

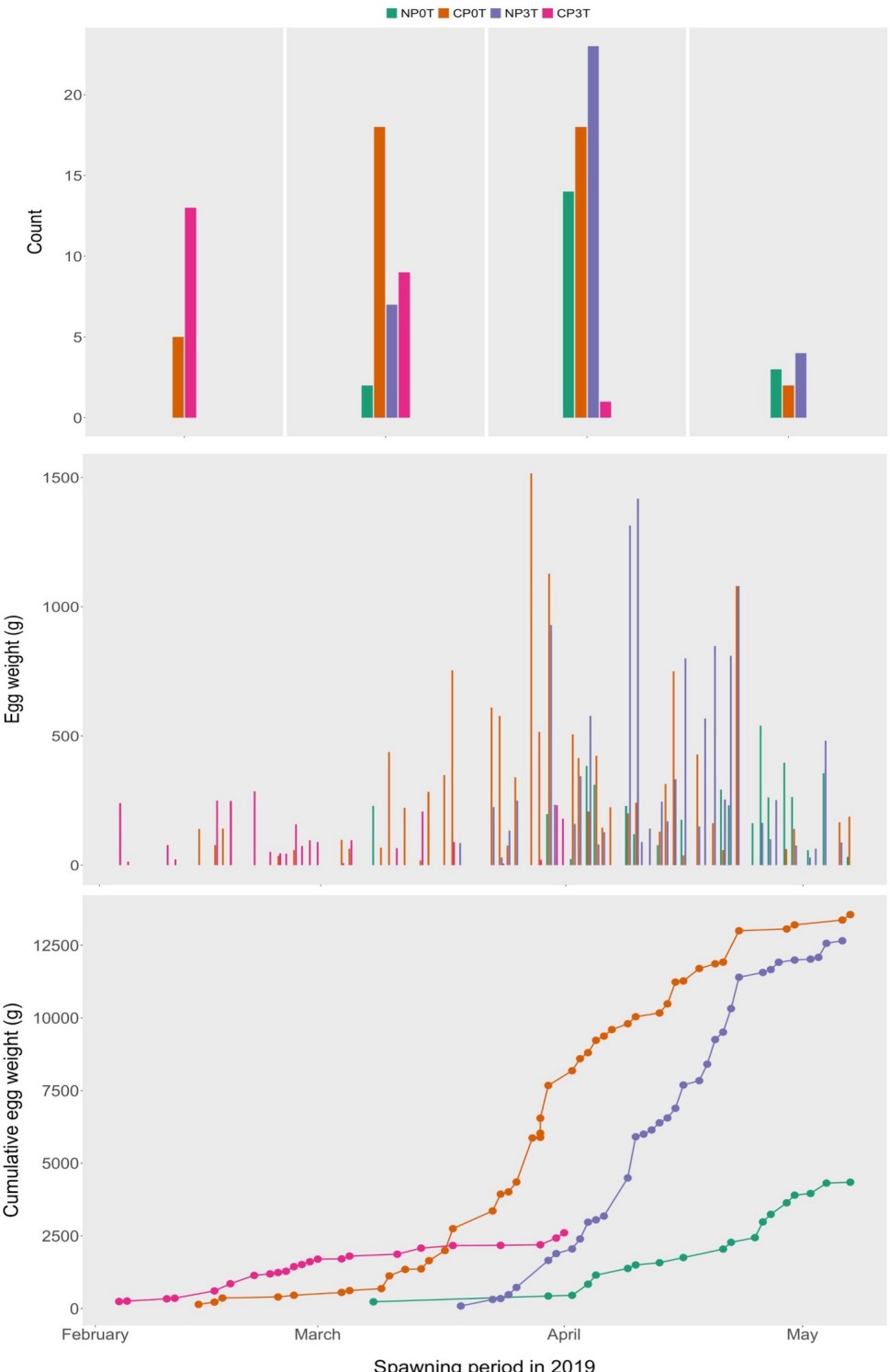

**Fig 8. Spawning counts and egg weights.** The number of spawns and weights of spawned eggs in lumpfish (Cyclopterus lumpus) females from different photothermal groups.

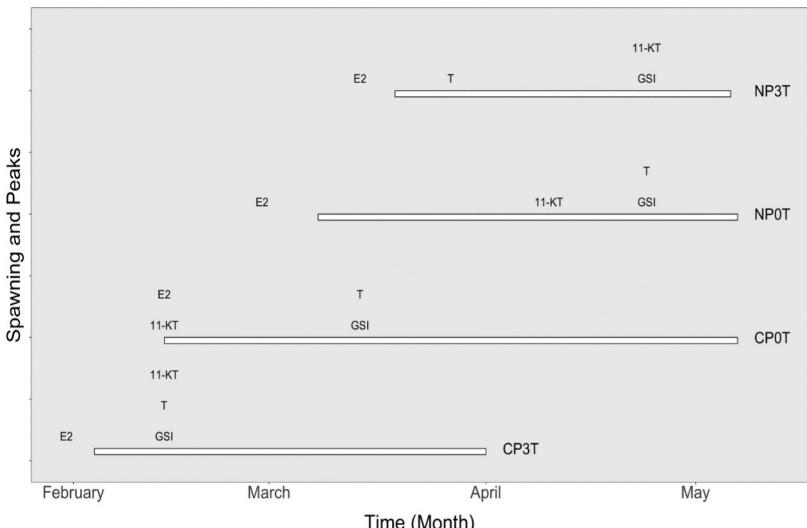

**Fig 9. Summary of the spawning and peak timings as obtained through sampling in 2019.** Peaks in GSI (gonadosomatic index), T (testosterone), 11-KT (11-ketotestosterone) and E2 (17β-estradiol) in different photothermal groups. The horizontal bars indicate the spawning period in each group, the differently positioned abbreviations indicate the peaks for each sexual maturation parameter.

in the compressed photoperiod. The low levels of GSI, testosterone (T), 11-ketotestosterone (11-KT), and 17β-estradiol (E2) before temperature elevation reflect the presence of only primary growth (PG) and secondary growth (SG) stages of ovarian development during this period. Furthermore, the earlier increase and decline of E2 compared to T and 11-KT demonstrate the crucial role of E2 in vitellogenesis [34, 35]. The long PG period, persisting for at least six months before the significant recruitment of SG is crucial for a complete synthesis of organelles and molecules necessary for later stages [36]. The SG period was also long, lasting five months before OM. However, we acknowledge that our sampling intervals may have missed the exact timing of oocyte recruitment because ovarian development is a continuous process. This aligns with available knowledge suggesting that lumpfish ovary development may require at least eight months for first spawning individuals, but the true duration remains unknown [33].

Before the temperature elevation in the CP, the significantly higher GSI, T and 11-KT observed on 3[rd] January 2019 suggest that the the ovary growth and androgen development were advanced in CP compared to NP. The higher recruitment of oocyte maturation, and non-significantly higher E2 levels also indicate that these parameters were advanced due to compression of the natural annual photoperiod. After temperature elevation, the effect of photoperiod was reflected by the changes in the sexual maturation parameters with fluctuations more pronounced in the compressed photoperiod. The fluctuations in both photoperiods could be attributed to the multiple batch spawning strategy in lumpfish [33, 37, 38]. This strategy suggests that different oocyte batches in the same ovary require varying levels of the sex steroids, while the spawning and recruitment of ready-to-spawn eggs contribute to fluctuations in GSI. The increased fluctuations in the CP group possibly resulted from the accelerating effect in this group, which advances sexual maturation events through earlier activation of the brain-pituitary (BPG) axis [39] which is central in the regulation of reproduction [13, 40]. In contrast, likely due to it's delaying effect, the NP group experienced fewer fluctuations. Due to the natural photoperiod´s delaying effect, a rather slower decline in E2 was observed, while

T, 11-KT and GSI increased. In agreement to our findings, advancement or compression of the natural photoperiods has been shown to alter the phases of GSI, and sex steroid developments in other species such as Atlantic cod (*Gadus morhua*) [41, 42] and Murray cod (*Maccullochella peelii peelii*) [35].

Ovarian development after temperature elevation demonstrated recruitment of more oocyte maturation (OM) and ovulation (OV) stages as females approached the spawning period. There were, however, no clear differences between the photoperiods apart from the comparison of OV recruitment between NP0T and NP3T from 27th March 2019 to 24th April 2019. This is likely due to the sampling intervals which mismatched with the continuity of ovarian development leading to spawning. Regarding spawning, in the compressed photoperiod, it was advanced by 1 month compared to the natural photoperiod. Similarly, in the female striped trumpeter (*Latris linaeta*), spawning was advanced by 1 and 4 months during consecutive spawning seasons following exposure to a 9-month compressed natural photoperiod [43]. Recent studies on lumpfish reported more predictable spawning and a peak in spawning under a compressed photoperiod compared to a natural photoperiod [9, 19].

Temperature elevation accelerated the changes in sexual maturation in the high temperature groups compared to their ambient temperature counterparts. The elevation of temperature caused higher levels, earlier peaks and declines of T, 11-KT and E2 in the compressed photoperiod, explaining why CP3T had the overall highest mean level of these sex steroids. It is believed that temperature influences the physiological aspects of the BPG axis to regulate ovary development [44, 45]. Therefore, the higher temperature led to further advancement of the BPG axis activation, as indicated by increased variations in sex steroids and greater energy accumulation in the lumpfish females from the high temperature groups, similar to observations in Atlantic salmon [46, 47]. The advancement of the BPG axis activation has been associated with temperature-induced stimulation of the brain regions integrating external and internal signals to regulate sexual maturation [13, 47, 48]. Consequently, temperature elevation accelerated the accumulation of sexual maturation resources earlier [46] compared to ambient temperature groups. Similarly, Morgan and colleagues found that in their natural environment, Atlantic cod displayed higher GSI in warm temperatures [49]. Earlier changes in T levels in Murray cod were observed when the temperature changes were also advanced [35]. Temperature elevation was responsible for the higher occurrence of OV in NP3T compared to NP0T in most of the sampling points. Similar dynamics possibly occurred between CP3T and CP0T, but due to advanced spawning in CP3T, imbalanced sampling, and potential mismatch of timing between oocyte recruitment and sampling, this observation was not possible. Thus, CP3T recruited ovulated eggs the earliest compared to the other groups.

Due to temperature elevation, CP3T spawned 1.5 weeks earlier than CP0T, which equates to 1.5 months earlier than in the natural photoperiod. Temperature elevation is argued here as the cause for earliest spawning within CP3T. Furthermore, the higher cumulative egg weight in NP3T than in NP0T, despite the earlier spawning in NP0T, indicates that temperature elevation still influences the final maturation, ovulation and subsequently, spawning. Additionally, we suggest that, without delayed temperature elevation, spawning in NP3T would have occurred earlier than in NP0T. In agreement with this, greater oocyte recruitment and earlier spawning have been linked to warmer temperatures in the Atlantic cod [50]. Also, Imsland and colleagues speculated a possible confounding effect of temperature on the spawning of lumpfish [19], a factor we have addressed in our study from the onset of oocyte maturation. On the other hand, the lower spawn weights, and numbers of spawned egg batches in CP3T may imply that elevating the temperature in addition to compressing the natural annual photoperiod may have affected the fecundity in these females differently from CP0T and NP3T. In

agreement to this, research has shown that different photoperiods and temperatures can cause alterations in fecundity as shown in the Atlantic cod [51, 52].

The presence of atretic follicles in fish ovaries is normal throughout the ovarian cycle [53]. The higher occurrence of atretic follicles before ovulation may be associated with the elimination of oocytes not ready for final maturation and ovulation [54]. Based on available energy levels in the females, eliminating portions of oocytes to sufficiently supply for the remaining oocytes through vitellogenesis to the subsequent stages [12] is necessary. Atretic follicles during the spawning period could be associated with the elimination of damaged or abnormal oocytes [53]. However, the observed atresia could also be due to furunculosis, which the fish were diagnosed with. This is possible due the high incidence of atresia at the same time of the disease diagnosis, and the later decrease possibly due to the successful medication. Since atresia above physiological rates can result from many factors [53], future studies on atresia in captivity can contribute to optimizing fecundity and reproductive success in female lumpfish.

Although not every sampling point included all four groups after temperature elevation, monitoring of spawning events enabled the creation of an arbitrary map illustrating peaks of the sexual maturation parameters in different groups. GSI peaked four weeks earlier in CP3T than in CP0T, and ten weeks earlier in CP3T than in both NP0T and NP3T. The peaks in T within CP3T were four, six and ten weeks earlier than in CP0T, NP3T and NP0T, respectively. For 11-KT in CP3T, the peaks aligned with those in CP0T and were eight and ten weeks earlier than in NP0T and NP3T, respectively. E2 in CP3T peaked to its highest two weeks earlier than in CP0T and four and six weeks earlier than in NP0T and NP3T, respectively (see Fig 9).

The temperature elevation was conducted at different maturation and photoperiod phases. This possibly contributed to the observed trends in the sexual maturation parameters between and within the photoperiods. In the compressed photoperiod, temperature elevation began at the onset of OM (3rd January 2019) when the fish were receiving 18 hours of light. In the natural photoperiod, temperature elevation began after the onset of OM, possibly with the OV already recruited (6th March 2019) when the fish were receiving 13 hours of light. Timing of temperature elevation is important since it can influence sexual maturation of fish differently in different photoperiod cycles [55]. Hence future studies on the influence of temperature elevation at different phases of the reproductive and photoperiodic cycles in lumpfish are necessary.

The findings of this study suggest that photothermal manipulations can be used to effectively control sexual maturation of lumpfish. Referring to the spawning which was advanced by up to 1.5 months through the compression of the natural photoperiod and elevation of temperature, the use of two distinct photoperiod and temperature could enable the production of up to four batches of lumpfish juveniles annually. However, further research is required to optimize these photoperiod and temperature manipulations. Such optimization could enhance the timing between batches originating from different photothermal groups. By adjusting the timing of photoperiod and temperature changes, and by testing photoperiods beyond those in this study, the predictability of delaying or accelerating sexual maturation could be improved. These strategies have the potential to enhance the efficiency of lumpfish farming operations. Furthermore, since the growth and sexual maturation responses vary under different photothermal regimes, the selection of desirable traits could be tailored to specific photoperiod and temperature settings, potentially enhancing the efficiency of the breeding program.

This study faced several limitations which may have influenced the findings. The initial fish density was determined based on the maximum carrying capacity of the tanks. Given the long-term nature of the experiment, the need to sacrifice fish and the frequent sampling requirements, smaller sample sizes were necessary. Although efforts were made to balance the study design, the reduced sample sizes may have contributed to the observed variations in the sexual maturation parameters. Additionally, challenges posed by atypical furunculosis led to

adjustments in the sampling routines, with different groups being sampled at varying intervals. As a result, some groups missed samples at specific time points, increasing the complexity of the analyses. The lack of consistent comparisons between groups at these time points may have limited valuable insights into the photothermal effects on sexual maturation. For future studies, careful consideration of sampling frequency and sample size is essential. Less frequent sampling may allow for larger sample sizes, and avoiding fish sacrifice with methods such as ultrasound [20] and catheterization [56] in conjunction with sex steroid assessments could improve the reliability of findings in a similar experimental setup.

Although our findings indicate that photothermal control can be used to achieve desired sexual maturation and spawning in lumpfish to meet annual demands, the subsequent effect on egg quality is not yet well understood. In a recent study [8], lumpfish broodstock were exposed to high temperature (14 ˚C) under low intensity 24-hour photoperiod for 11 weeks and 4 days, resulting in the production of non-viable eggs (to the eyed stage). Furthermore, anecdotal evidence suggests that lumpfish caught from the wild during autumn in mid-Norway produce poorer eggs than those caught during the winter and spring seasons [8]. Despite the recommended thermal optimum of $< 10$˚C [8], the preceding effects of different photoperiods which may require different thermal optima, have not yet been studied.

While lice eating ability is considered as the most important trait for improvement through selective breeding [2, 57], optimizing growth and sexual maturation with minimal individual variations is also paramount. Considering the overview of growth and sexual maturation in the lumpfish females from the present study, we believe that incorporating these parameters into selective breeding programs will contribute to developing females synchronized to photo-thermal manipulations with minimal individual variations. This optimization can maximize the use of facilities by ensuring the size of better-performing female stocks in differently pro-grammed breeding cycles in lumpfish hatcheries.

## Conclusion

Although statistical significance was not consistently observed in several between-group comparisons, our findings indicate that compressing the natural photoperiod and elevating the ambient temperature by 3˚C advanced sexual maturation and spawning in lumpfish females. Compressing the natural photoperiod advanced ovarian development, resulting in higher GSI, T, 11-KT and E2 levels. Temperature elevation caused advanced timing and higher recruitment of ovulation, improved GSI, T, 11-KT and E2 levels, and led to their earlier declines. Spawning was advanced under the compressed photoperiod by 1 month, which improved to 1.5 months with temperature elevation compared to the natural photoperiod. These findings imply that breeding programs and commercial lumpfish farms can benefit from the predictability of advancement and synchronization of sexual maturation and spawning. Moreover, using different photothermal regimes to either delay or advance sexual maturation in different groups of fish allows for multiple spawning cycles per year, leading to more efficient use of the production facilities. Despite the potential success of manipulating both photoperiod and temperature for control of lumpfish reproduction in captivity, the timing of temperature elevation, and the adjacent effects on fecundity and egg quality necessitate further studies.

## Supporting information

**S1 Fig. Temperature and dissolved oxygen levels in different groups throughout the experimental period.**
(PDF)

**S2 Fig. Photoperiod and temperature effects on condition factor.** Condition factors of lumpfish females from different photoperiods (until 03-Jan-19) and different photoperiod and temperature combinations (from 03-Jan-19). NP0T: natural photoperiod at ambient temperature, CP0T: compressed photoperiod at ambient temperature, NP3T: natural photoperiod at elevated temperature, CP3T: compressed photoperiod at ambient temperature. The red rectangle on 03-Jan-2019 indicates the start of temperature elevation, and the red vertical line separates the periods before and after temperature elevation. Values are mean ± S.E.M and statistical significances between groups at a sampling point are indicated by "*" or "ns". Statistical differences between sampling points are graphically displayed in supporting information on S8–14 Figs. For trend visualization, refer to (S6 Fig), which presents the data as a line graph.
(PDF)

**S3 Fig. Mortality and cumulative mortality of lumpfish.** Number of dead fish (left y aixs) and cumulative number of dead fish (right y axis) in different photothermal regimes. T1, T2, T3 and T4 represent Tank 1, Tank 2, Tank 3, and Tank 4, respectively.
(PDF)

**S4 Fig. Line graph showing trends of mean values of body weight (top) and gonadosomatic index (bottom) from different photothermal regimes.** Body weights and gonadosomatic indices of lumpfish females from different photoperiods (until 03-Jan-19), and different photoperiod and temperature combinations (from 03-Jan-19). NP0T: natural photoperiod at ambient temperature, CP0T: compressed photoperiod at ambient temperature, NP3T: natural photoperiod at elevated temperature, CP3T: compressed photoperiod at ambient temperature.
(PDF)

**S5 Fig. Line graph showing trends of mean values of sex steroids from different photothermal regimes.** Blood plasma levels of testosterone (top), 11-ketotestosterone (middle) and 17β-estradiol (bottom) in lumpfish females from different photoperiods (until 03-Jan-19), and different photoperiod and temperature combinations (from 03-Jan-19). NP0T: natural photoperiod at ambient temperature, CP0T: compressed photoperiod at ambient temperature, NP3T: natural photoperiod at elevated temperature, CP3T: compressed photoperiod at ambient temperature.
(PDF)

**S6 Fig. Line graph showing trends of mean values of condition factor from different photothermal regimes.** Condition factors of lumpfish females from different photoperiods (until 03-Jan-19) and different photoperiod and temperature combinations (from 03-Jan-19). NP0T: natural photoperiod at ambient temperature, CP0T: compressed photoperiod at ambient temperature, NP3T: natural photoperiod at elevated temperature, CP3T: compressed photoperiod at ambient temperature.
(PDF)

**S7 Fig. Oogenesis in lumpfish visualized through histology.** 10 stages were identified: pn, perinucleolus; ca, cortical alveoli; od, oil droplet; py, primary yolk; sy, secondary yolk; ty, tertiary yolk; mat, maturing; matd, matured oocytes and ov, ovulating oocytes; pov, post-ovulatory eggs. Scale bars are shown in the Figure.
(PDF)

**S8 Fig. Pairwise comparisons of body weight and condition factor changes over time before temperature elevation.** There are separate panels for each group, representing estimated mean differences between sampling points. Within each panel, the contrasts between

sampling points suggesting temporal variations in the body weight and condition factor are
shown. Contrasts where blue bars do not cross the zero exhibit statistically significant differences, indicating varying impacts of time on body weight and condition factor in the different
groups.
(PDF)

**S9 Fig. Pairwise comparisons of gonadosomatic index and testosterone changes over time
before temperature elevation.** There are separate panels for each group, representing estimated mean differences between sampling points. Within each panel, the contrasts between
sampling points suggesting temporal variations in gonadosomatic index and testosterone are
shown. Contrasts where blue bars do not cross the zero exhibit statistically significant differences, indicating varying impacts of time on gonadosomatic index and testosterone in the different groups.
(PDF)

**S10 Fig. Pairwise comparisons of 11-ketotestosterone and 17β-estradiol changes over time
before temperature elevation.** There are separate panels for each group, representing estimated mean differences between sampling points. Within each panel, the contrasts between
sampling points suggesting temporal variations in the 11-ketotestosterone and 17β-estradiol
are shown. Contrasts where blue bars do not cross the zero exhibit statistically significant differences, indicating varying impacts of time on 11-ketotestosterone and 17β-estradiol in the
different groups.
(PDF)

**S11 Fig. Pairwise comparisons of body weight and condition factor changes over time after
temperature elevation.** There are separate panels for each group, representing estimated
mean differences between sampling points. Within each panel, the contrasts between sampling
points suggesting temporal variations in the body weight and condition factor are shown. Contrasts where blue bars do not cross the zero exhibit statistically significant differences, indicating varying impacts of time on body weight and condition.
(PDF)

**S12 Fig. Pairwise comparisons of gonadosomatic index changes over time after temperature elevation.** There are separate panels for each group, representing estimated mean differences between sampling points after temperature elevation. Within each panel, the contrasts
between sampling points suggesting temporal variations in the gonadosomatic index are
shown. Contrasts where blue bars do not cross the zero exhibit statistically significant differences, indicating varying impacts of time on gonadosomatic index in the different groups.
(PDF)

**S13 Fig. Pairwise comparisons of testosterone and 11-ketotestosterone changes over time
after temperature elevation.** There are separate panels for each group, representing estimated
mean differences between sampling points. Within each panel, the contrasts between sampling
points suggesting temporal variations in the testosterone and 11-ketotestosterone are shown.
Contrasts where blue bars do not cross the zero exhibit statistically significant differences,
indicating varying impacts of time on testosterone and 11-ketotestosterone in the different
groups.
(PDF)

**S14 Fig. Pairwise comparisons of 17β-estradiol changes over time after temperature elevation.** There are separate panels for each group, representing estimated mean differences
between sampling points. Within each panel, the contrasts between sampling points suggesting

temporal variations in 17β-estradiol are shown. Contrasts where blue bars do not cross the zero exhibit statistically significant differences, indicating varying impacts of time on 17β-estradiol in the different groups.
(PDF)

**S1 Table. Sampling points and sample sizes for each group during the experiment.** The blue shaded region is the sampling period before temperature elevation, the orange shaded region is the sampling period after temperature elevation. In seven sampling points after temperature elevation, one to two groups were not sampled.
(PDF)

**S2 Table. Mean temperature and oxygen levels.** The values for NP3T and CP3T are specific to the period after temperature elevation. Values are mean ± SD.
(PDF)

**S1 Data.**
(ZIP)

## Acknowledgments

The authors want to thank Senior engineer Tora Bardal of the Department of Biology, Norwegian University of Science and Technology, for providing support in terms of resources, teaching and advice during tissue processing, imaging, and assessment of histology images.

## Author Contributions

**Conceptualization:** Maren Mommens, Helge Tveiten, Jonna Tomkiewicz, Elin Kjørsvik, Velmurugu Puvanendran.

**Data curation:** Frank Thomas Mlingi, Erik Burgerhout, Helge Tveiten, Elin Kjørsvik, Velmurugu Puvanendran.

**Formal analysis:** Frank Thomas Mlingi.

**Funding acquisition:** Maren Mommens, Helge Tveiten, Jonna Tomkiewicz, Elin Kjørsvik, Velmurugu Puvanendran.

**Investigation:** Frank Thomas Mlingi, Erik Burgerhout, Maren Mommens, Helge Tveiten, Elin Kjørsvik, Velmurugu Puvanendran.

**Methodology:** Frank Thomas Mlingi, Erik Burgerhout, Maren Mommens, Helge Tveiten, Jonna Tomkiewicz, Elin Kjørsvik, Velmurugu Puvanendran.

**Project administration:** Velmurugu Puvanendran.

**Resources:** Erik Burgerhout, Helge Tveiten, Jonna Tomkiewicz, Elin Kjørsvik, Velmurugu Puvanendran.

**Supervision:** Erik Burgerhout, Helge Tveiten, Jonna Tomkiewicz, Elin Kjørsvik, Velmurugu Puvanendran.

**Validation:** Helge Tveiten, Jonna Tomkiewicz, Elin Kjørsvik, Velmurugu Puvanendran.

**Visualization:** Frank Thomas Mlingi.

**Writing – original draft:** Frank Thomas Mlingi.

**Writing – review & editing:** Frank Thomas Mlingi, Erik Burgerhout, Maren Mommens, Helge Tveiten, Jonna Tomkiewicz, Elin Kjørsvik, Velmurugu Puvanendran.

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
