## [Decision Letter · Decision Letter 0]

26 Aug 2024

PONE-D-24-25185Reproductive performance of lumpfish (Cyclopterus lumpus, L. 1758) females: Effects of integrated photoperiod and temperature manipulations on sexual maturation and spawningPLOS ONE

Dear Dr. Mlingi,

Thank you for submitting your manuscript to PLOS ONE. After careful consideration, we feel that it has merit but does not fully meet PLOS ONE’s publication criteria as it currently stands. Therefore, we invite you to submit a revised version of the manuscript that addresses the points raised during the review process.

We look forward to receiving your revised manuscript.

Kind regards,

A. K. Shakur Ahammad, PhD

Academic Editor

PLOS ONE

Journal Requirements:

Additional Editor Comments:

Thank you for submitting your manuscript to PLOS ONE. Based on the review, a "minor revision" is requested for the following areas:

1. Introduction: Provide more detail on current challenges in lumpfish breeding.

2. Materials and Methods: Consider adding a flowchart for clarity.

3. Results: Include additional graphs and provide more context on significant findings.

4. Discussion and Conclusion: Highlight practical applications for aquaculture and discuss study limitations and future research.

We look forward to receiving your revised manuscript.

Reviewers' comments:

Reviewer's Responses to Questions

**Comments to the Author**

1. Is the manuscript technically sound, and do the data support the conclusions?

Reviewer #1: Yes

Reviewer #2: Yes

2. Has the statistical analysis been performed appropriately and rigorously? 

Reviewer #1: Yes

Reviewer #2: Yes

3. Have the authors made all data underlying the findings in their manuscript fully available?

Reviewer #1: Yes

Reviewer #2: Yes

4. Is the manuscript presented in an intelligible fashion and written in standard English?

Reviewer #1: Yes

Reviewer #2: Yes

5. Review Comments to the Author

Reviewer #1: Reviewer Feedback

Title: Reproductive performance of lumpfish (Cyclopterus lumpus, L. 1758) females: Effects

of integrated photoperiod and temperature manipulations on sexual maturation and

spawning

Manuscript number: PONE-D-24-25185

Overall Recommendations: Minor Revision!

Section wise feedback

1.Title and Abstract:

oThe title is clear and accurately reflects the content of the manuscript.

oThe abstract provides a concise summary of the study's aims, methods, results, and conclusions. It effectively highlights the significance of photoperiod and temperature manipulations on lumpfish reproduction.

2.Introduction:

oThe introduction provides a comprehensive background on the importance of lumpfish in aquaculture and the need for controlled reproduction. However, it could benefit from a more detailed explanation of the current challenges in lumpfish breeding and how this study addresses those gaps.

3.Materials and Methods:

oThe experimental design is robust, and the methods are described in detail, allowing for reproducibility. The methodology section is comprehensive, but it would be helpful to include a flowchart summarizing the experimental design, treatments, and key measurements. This visual aid can enhance clarity for readers.

4.Results:

oThe results are detailed and well-organized, but the inclusion of more graphical representations (e.g., line charts or bar graphs) could make complex data trends more accessible.

oWhen discussing significant findings, consider adding more context about the biological significance of these results. For example, explain why certain hormone levels fluctuated and how this impacts lumpfish reproduction.

oConsider providing more detailed explanations for the observed fluctuations in body weight and condition factor after temperature elevation.

5.Discussion:

oThe discussion effectively interprets the results in the context of existing literature.

oThe discussion is thorough, but it would benefit from a clearer connection between the study’s findings and their practical implications for lumpfish aquaculture. How can these results be applied to improve breeding practices?

oAddress any limitations of the study more explicitly and suggest areas for future research to provide a balanced perspective.

6.Conclusion:

Highlighting how these findings can be implemented in aquaculture settings would emphasize the study's relevance.

Reviewer #2: General comments

The authors have reported the combined effects of photoperiod and temperature manipulations on sexual maturation and spawning in lumpfish females by exposing to natural and compressed annual photoperiods, with subsequent temperature elevation.

The authors have reported that compressing the natural photoperiod causes a clear increase and decrease in gonado somatic index (GSI), testosterone (T), 11-ketotestosterone (11-KT) and 17β-estradiol (E2) towards and during the spawning period. The authors have reported that spawning could be advanced by 1 month in the compressed photoperiod compared to the natural photoperiod; and temperature elevation could advance spawning by 1.5 months in the compressed photoperiod compared to the natural photoperiod.

They have concluded that compressing the natural photoperiod and elevating temperature can advance sexual maturation and spawning in lumpfish females indicating a successful control of sexual maturation for year-round production of lumpfish juveniles.

The study is well designed, the methodology is adequate to observe the effects of the variables, the results are sufficient to address the objectives.

6. PLOS authors have the option to publish the peer review history of their article (what does this mean?). If published, this will include your full peer review and any attached files.

Reviewer #1: No

Reviewer #2: No

---

## [Author Response · Author response to Decision Letter 0]

23 Sep 2024

Reviewer Feedback

Title: Reproductive performance of lumpfish (Cyclopterus lumpus, L. 1758) females: Effects

of integrated photoperiod and temperature manipulations on sexual maturation and

spawning 

Manuscript number: PONE-D-24-25185 

Overall Recommendations: Minor Revision!

Section wise feedback

1. Title and Abstract: 

 -The title is clear and accurately reflects the content of the manuscript.

 -The abstract provides a concise summary of the study's aims, methods, results, and conclusions. It effectively highlights the significance of photoperiod and 

 temperature manipulations on lumpfish reproduction.

2. Introduction:

 -The introduction provides a comprehensive background on the importance of lumpfish in aquaculture and the need for controlled reproduction. However, it could 

 benefit from a more detailed explanation of the current challenges in lumpfish breeding and how this study addresses those gaps. 

 Thank you very much for your suggestion. In response, we have added a paragraph explaining several traits and the associated challenges in lumpfish as cleaner 

 fish. Additionally, we have explained how breeding programs aim to address these challenges and how our study can contribute to successful development of 

 such programs. The paragraph has been positioned to ensure a smooth transition to the subsequent sections, which focus on the specific aims of this study.

3. Materials and Methods:

 -The experimental design is robust, and the methods are described in detail, allowing for reproducibility. The methodology section is comprehensive, but it would 

 be helpful to include a flowchart summarizing the experimental design, treatments, and key measurements. This visual aid can enhance clarity for readers.

 Thank you very much for your suggestion. In response, we have included a flowchart that outlines the experiment, sampling, and data collection process as a 

 figure.

4. Results:

 -The results are detailed and well-organized, but the inclusion of more graphical representations (e.g., line charts or bar graphs) could make complex data trends 

 more accessible.

 Thank you very much for your suggestion. We acknowledge that, due to the nature of the experimental design and some unexpected events, the data trends are 

 indeed complex. To aid in understanding these trends, we included line charts in the initial submission (Figures 5 and 6). In this revision, we have added 

 supplementary line graphs for body weight, condition factor, gonadosomatic index and sex steroids presenting only the mean values. These additional graphs are 

 included in the supplementary materials to avoid what we felt could be duplicate data visualizations. We hope this approach will enhance understanding for 

 interested readers.

 -When discussing significant findings, consider adding more context about the biological significance of these results. For example, explain why certain hormone 

 levels fluctuated and how this impacts lumpfish reproduction. 

 Thank you for your feedback. We have added possible explanations regarding the biological significance of variations in the sexual maturation indicators to 

 enhance relevant discussions in the discussion section, particularly in relation to photoperiod and temperature. 

 -Consider providing more detailed explanations for the observed fluctuations in body weight and condition factor after temperature elevation.

 Thank you for your suggestion. We have added more explanations regarding the fluctuations in body weight and condition factor in the respective sections.

5. Discussion:

 -The discussion effectively interprets the results in the context of existing literature.

 -The discussion is thorough, but it would benefit from a clearer connection between the study’s findings and their practical implications for lumpfish aquaculture. 

 -How can these results be applied to improve breeding practices?

 Thank you for your suggestion. A paragraph highlighting the practical implications of the study’s findings on lumpfish aquaculture has been added. This paragraph 

 discusses how the findings related to photothermal manipulations can be implemented and optimized in lumpfish juvenile production. Furthermore, we have 

 addressed the need for additional research aimed at optimizing these photothermal manipulations and enhancing the efficiency of breeding programs. 

 -Address any limitations of the study more explicitly and suggest areas for future research to provide a balanced perspective.

 Thank you for your suggestion. We have highlighted limitations of the study related to its nature, small sample sizes, fish sacrifice and atypical furunculosis 

 challenge in the discussion. Additionally, we have suggested considerations for future studies regarding sample size and sampling frequency, as well as replacing 

 fish sacrifice with less invasive methods. This paragraph has been positioned towards the end of the discussion. 

6. Conclusion: 

 -Highlighting how these findings can be implemented in aquaculture settings would emphasize the study's relevance.

 Thank you for your feedback. We have highlighted how the findings can be implemented in aquaculture settings. Additionally, we highlight how breeding 

 programs and commercial lumpfish farms may benefit from the photothermal manipulations.

---

## [Editor Report · Decision Letter 1]

25 Sep 2024

Reproductive performance of lumpfish (Cyclopterus lumpus, L. 1758) females: Effects of integrated photoperiod and temperature manipulations on sexual maturation and spawning

PONE-D-24-25185R1

Dear Dr. Mlingi,

We’re pleased to inform you that your manuscript has been judged scientifically suitable for publication and will be formally accepted for publication once it meets all outstanding technical requirements.

Kind regards,

A. K. Shakur Ahammad, PhD

Academic Editor

PLOS ONE
---

## [Editor Report · Acceptance letter]

30 Sep 2024

PONE-D-24-25185R1 

PLOS ONE

Dear Dr. Mlingi, 

I'm pleased to inform you that your manuscript has been deemed suitable for publication in PLOS ONE. Congratulations! Your manuscript is now being handed over to our production team.

Kind regards, 

on behalf of

Dr. A. K. Shakur Ahammad 

Academic Editor

PLOS ONE